# Characterization of the Fluidity of the Ultrasonic Plasticized Polymer Melt by Spiral Flow Testing under Micro-Scale

**DOI:** 10.3390/polym11020357

**Published:** 2019-02-18

**Authors:** Bingyan Jiang, Yang Zou, Tao Liu, Wangqing Wu

**Affiliations:** 1School of Mechanical and Electrical Engineering, Lushan South Road 932, Changsha 410083, Hunan, China; jby@csu.edu.cn (B.J.); yangzoucsu@163.com (Y.Z.); csuliutao@126.com (T.L.); 2State Key Laboratory of High Performance Complex Manufacturing, Central South University, Lushan South Road 932, Changsha 410083, Hunan, China

**Keywords:** ultrasonic plasticizing, microinjection molding, spiral flow testing, fluidity

## Abstract

The fluidity of a molten polymer plasticized by ultrasonic vibration was characterized by spiral flow testing based on an Archimedes spiral mold with microchannels. Mold inserts with various channel depths from 250 to 750 µm were designed and fabricated to represent the size effect under micro-scale. The effect of ultrasonic plasticizing parameters and the mold temperature on the flow length was studied to determine the rheological nature of polymers and control parameters. The results showed that the flow length decreased with reduced channel depth due to the size effect. By increasing ultrasonic amplitude, ultrasonic action time, plasticizing pressure, and mold temperature, the flow length could be significantly increased for both the amorphous polymer polymethyl methacrylate (PMMA) and the semi-crystalline polymers polypropylene (PP) and polyamide 66 (PA66). The enhanced fluidity of the ultrasonic plasticized polymer melt could be attributed to the significantly reduced shear viscosity.

## 1. Introduction

Ultrasonic microinjection molding has become an attractive alternative molding technology for polymeric micro components. Instead of traditional heater band and screw pressing and shearing, high-frequency periodic mechanical vibration energy works as a plasticizing agent in ultrasonic microinjection molding [1]. This vibration-induced inside-out heat generation plasticizing is much more energy efficient than the traditional outside-in heat conduction. In addition, the plasticized molten material is further influenced by the ultrasonic agitation effect, leading to a reduced melt viscosity and enhanced cavity filling performance. Ultrasonic microinjection molding has gained extensive attention in recent years due to its great potential to reduce energy consumption, increase materials utilization, and enhance molding performance [2,3,4,5,6,7,8,9,10,11,12,13,14].

Previously, we reported our results regarding the polymer plastification mechanism, specifically, interfacial friction heating [15] and volumetric viscoelastic heating [16] of polymer pellets. This study focused on the influence of the ultrasonic agitation effect on the plasticized molten material. In ultrasonic-assisted extrusion/injection molding, high-frequency periodic ultrasonic vibrations can shear the polymer melt, causing the entangled polymer chains to unwind and align along the melt flow direction, reducing the viscosity of the polymer melt [17,18,19,20]. Ultrasound has both physical and chemical effects on the melt viscosity of polypropylene according to Chen et al. [21]. The physical effects include an increase of molecular chain kinematic activity and the promotion of unwrapping, while the chemical effects include chemical bond cleavage and molecular weight reductions.

In ultrasonic microinjection molding, the plasticized molten materials can be significantly different from the ones in traditional extrusion/injection molding. Due to the inside-out heat generation plasticizing, the melt temperature in ultrasonic microinjection molding is non-deterministic and depends on the ultrasonic plasticizing conditions. One cannot simply specify an indicator such as the melt flow index (MFI) to represent the flow properties of the melt plasticized by ultrasound. Moreover, the ultrasonic agitation effect starts from the plasticization of the polymer pellets, which is substantially extended in comparison with traditional extrusion/injection molding. Therefore, the fluidity of the polymer melt can be further altered by the prolonged ultrasonic irradiation. This was confirmed by our initial MFI test in which a fully ultrasonic plasticized and consolidated polypropylene (PP) cylinder was used [22]. It was found that an increased ultrasonic amplitude and plasticizing pressure are beneficial to improve the polymer melt fluidity. Similar results were obtained by Michaeli and Opfermann [3] and Sacristán et al. [6]. Their results show that the filling performance of the ultrasonic plasticized polymer melt can be significantly influenced by the ultrasonic amplitude and plasticizing pressure.

To characterize the fluidity of polymer melt directly after ultrasonic plasticization, it is necessary that the testing method can approximately simulate the ultrasonic microinjection molding process as much as possible. Unfortunately, there is no available technology to this end, since commercial testing equipment such as high-pressure capillary rheometers, rotary rheometers, and MFI apparatus are all based on the outside-in heat conduction plasticizing concept. Therefore, in this work, a spiral flow testing based on an Archimedes spiral mold [23,24,25] featured with micro-channels is adapted especially for ultrasonic microinjection molding. The influence of ultrasonic amplitude, plasticizing pressure, ultrasonic action time, and micro-scale effects on the fluidity of the polymer melt was investigated via single-factor analysis. The effects of various material types were considered as well. The obtained results are intended to provide a reference for mold design and process development for ultrasonic microinjection molding.

## 2. Experimentation

### 2.1. Materials

Three kinds of material were used in this study, that is, polymethyl methacrylate (PMMA, TF8, Mitsubishi Chemical Holdings Group, minato-ku, Tokyo, Japan), polypropylene (PP, K1011, Formosa Chemicals & Fiber Corporation, Yunlin County, Taiwan, China) and polyamide 66 (PA66, Zytel101NC010, DuPont Corporation, Washington, VA, USA). Their material properties are as shown in Table 1.

### 2.2. Equipment

In-house-developed ultrasonic microinjection molding equipment and an Archimedes spiral mold were used in this study, as shown in Figure 1a [26]. The ultrasonic frequency was 20 kHz and the amplitude ranged from 28 to 52 μm. The Archimedes spiral channel was on the upper surface of the mold, as shown in Figure 1b. The equation of the spiral was ρ=2θ/π with 4π≤θ≤9π, where ρ is polar path, and θ is polar angle.

### 2.3. Methodology

Single-factor experiments were designed to investigate the influence of the process parameters on the fluidity of the polymer melt in ultrasonic microinjection molding. In addition to the ultrasonic amplitude, the ultrasonic action time, the plasticizing pressure, and the mold temperature, the studied process parameters also included the holding time and the holding pressure. To determine whether the holding phase had an influence on the cavity filling, the latter two process parameters were defined to characterize the holding phase which was added directly after stopping the ultrasonic vibration. The values of each process parameter are given in Table 2. Value 3 is the reference of each process parameter. When the value of the studied process parameter was changed, the value of the others were held constant at value 3. The filling length could be calculated by integrating the spiral equation. To study the influence of micro-scale effect on the fluidity of the polymer melt, three spiral channels with thicknesses of 750, 500, 250 μm and a width of 1500 μm were prepared, named as Mold I, Mold II, and Mold III, respectively.

## 3. Results and Discussion

### 3.1. The Influence of Ultrasonic Amplitude

The influence of ultrasonic amplitude on the filling length of the polymer melt is shown in Figure 2. When the ultrasonic amplitude was 32 μm, PMMA could be plasticized and injected with a filling length of 11.3 mm, but PP and PA66 were not completely plasticized and the length was 0. As can be seen from Table 1, the melting point of PMMA was the lowest, so the energy required for plasticization was smaller than that of PP and PA66. When the frequency is constant, the ultrasonic energy per unit time is proportional to the square of the amplitude. That means the ultrasonic wave with amplitude of 32 μm was enough to plasticize PMMA, but not enough to plasticize PP and PA66. When the ultrasonic amplitude reached 36 μm, the filling lengths of PP and PA66 were 23 and 25.1 mm, respectively. When the ultrasonic amplitude reached 40 μm, the polymers in the chamber could be completely plasticized. The filling lengths of polymers increased with increasing ultrasonic amplitude. When the ultrasonic amplitude exceeded 40 μm, the increase of the filling length began to slow down. The relationship between the filling lengths under the same conditions was: *L*_PA66_ > *L*_PP_ > *L*_PMMA_, which indicates that under the same conditions, the melt fluidity of PA66 was better than that of PP and PMMA, and PMMA had the worst melt fluidity. This can be attributed to the intrinsic flow characteristics of the three materials as well as the acoustic impedance of the material. The acoustic impedance reflects the ability of the material to consume sonic energy. The greater the acoustic impedance, the greater the attenuation of ultrasound, thus reducing the effect of ultrasound on flow performance. As can be seen from Table 1, the acoustic impedance of PMMA is higher than that of PP and PA66, so the ultrasonic plasticizing had the lowest influence on the melt fluidity of PMMA.

### 3.2. The Influence of Ultrasonic Action Time

The influence of ultrasonic action time on the filling length of the polymer melt is shown in Figure 3. Ultrasonic microinjection molding can be a highly energy-efficient and time-saving process. Usually, the polymer pellets can be plasticized and molded within a few seconds. For example, the PMMA was filled with 10.9 mm at 2 s, as shown in Figure 3. When the ultrasonic action time was increased to 4 s, PP and PA66 were filled with 12.5 and 29.5 mm, respectively. This could be ascribed to the fact that the semi-crystalline polymers need more energy to plasticize. When the ultrasonic action time was extended from 2 to 10 s, the PMMA filling length was increased by 2.8 times. From 4 to 10 s, the PP filling length was increased by 3.6 times. From 4 to 10 s, the PA66 fill length was increased by 1.7 times. As the ultrasonic action time exceeded 6 s, the increase of the filling length of the three materials began to slow down. This could be related to the heat generation mechanism during ultrasonic plasticizing [1,15,16,27]. The ultrasonic plasticizing heat generation mainly includes interfacial frictional heat generation and volumetric viscoelastic heat generation. When the ultrasound is turned on, the polymer pellets are compressed and experience a fierce interfacial friction. The temperature at the interface of the polymer pellets increases sharply. The contact interface is ablated in a very short time, and the interfacial frictional heat generation weakens rapidly as well. Then, viscoelastic heat generation dominates and becomes the main heat source. As the polymer melts, the viscoelastic heat generation rate also begins to decrease, but the heat generation still maintains the polymer in a molten state. In the early stage of plasticization, the heat generation rate is high, and the polymer temperature increases rapidly, leading to a sharp increase of the polymer filling length. In the later plasticization period, the heat generation rate is reduced, and the polymer temperature is gradually stabilized. However, the shearing effect of ultrasonic vibration on the melt still exists, so the filling length in the later plasticization period increases but the increment becomes smaller.

### 3.3. The Influence of Plasticizing Pressure

Plasticizing pressure refers to the pressure of the ultrasonic tool acting directly on the polymer pellets, and its effect on the filling of the three polymers is shown in Figure 4. When the pressure was 10 MPa, the filling lengths of PMMA, PP, and PA66 were 17.1, 33.2, and 36.4 mm, respectively. When the pressure was increased to 18 MPa, the filling lengths reached 38.9, 61.7, and 72.9 mm, respectively. The plasticizing pressure has a direct influence on the heat generation. The increase of pressure causes larger deformation of the polymer pellets and a greater contact area, so the efficiency of friction heating is increased as well. The load on the end face of the ultrasonic tool increases with the plasticizing pressure, leading to an increased energy input to the polymer by the ultrasonic vibration system.

### 3.4. The Influence of the Holding Process

In the holding stage of injection molding, the melted polymer continues to fill the mold cavity under the holding pressure. The holding process mainly plays the role of melt compensation, and can prevent melt flow back to the channel. In ultrasonic microinjection molding, the start time of the holding process is the time when the ultrasonic action is terminated. Therefore, the holding process has no significant influence on the plasticizing, but on the filling of the polymer melt. As shown in Figure 5a, when the holding time was extended from 2 to 6 s, the filling lengths of PMMA, PP, and PA66 were increased by 24.3%, 21.1%, and 24.1%, respectively. When the holding time was extended from 6 to 10 s, the filling lengths of the polymers increased by 10.2%, 5%, and 9.9%, respectively. There is no energy input during the holding stage, and the polymer melt starts to cool down. The holding process had a larger influence at the initial 2–6 s. After that, the polymer cools down and solidifies, so the prolonged holding had little influence. Figure 5b shows how the fill length varied with holding pressure. It can be seen that the filling length increased approximately linearly with the increase of holding pressure.

### 3.5. The Influence of Mold Temperature

The filling of the polymer melt in the cavity also depends on the mold temperature, due to the temperature difference between the mold insert and the polymer melt. When the polymer melt flows into the spiral channel, the energy of the melt is transferred to the mold insert via heat conduction [28,29], resulting in a decreased melt temperature and therefore a limited flowability. An increased mold temperature can effectively delay the energy loss of the polymer melt and increase the filling length. As shown in Figure 6, when the mold temperature increased from 40 to 80 °C, the filling lengths of PMMA, PP, and PA66 increased by 200%, 95%, and 61%, respectively, and the increase in mold temperature had the greatest impact on the filling of PMMA.

### 3.6. The Influence of the Size Effect

The influence of the channel thickness on the filling length of the polymer melt is shown in Figure 7. In general, the filling length increases with the mold temperature, as illustrated in Figure 6. However, the channel thickness has a significant influence on the filling length of the polymer melt. Comparing the filling results of the three molds, it can be found that the polymer melt in Mold I had the longest filling length and the largest increment. Following Mold II, the polymer melt in Mold III had the shortest filling length and the smallest increment. In microinjection molding, the surface-to-volume ratio increases with reduced channel thickness. The heat of the polymer melt can be rapidly transferred to the mold, leading to melt solidification and an increasing filling resistance. Increasing the mold temperature can slow down the cooling of the polymer melt, resulting in a longer filling length. However, in the case of Mold III, even if the mold temperature was increased, the increment of the filling length was limited.

Flow ratio refers to the ratio of the greatest filling length to the channel thickness when the melt flows in the mold under a certain injection pressure. The experimental results as indicated in Figure 7 were converted into flow ratios as shown in Figure 8. The filling lengths of polymers in different molds presented different behavior. Except for 80 °C, the flow ratios of PMMA and PP in Mold III were the largest, followed by Mold II, and the minimum flow ratio was in Mold I. The flow ratio of PA66 in Mold III was similar to the filling length in Mold I. The results show that the ultrasonic plasticized polymer also had good filling properties in the microcavity. Although the absolute filling length of the polymer in the small-sized flow channel was not as large as in the large-sized flow channel, it was sufficient for precision injection molding of small-sized parts. The results prove that the ultrasonic plasticization can be used in high-aspect ratio cavity filling.

## 4. Conclusions

Ultrasonic plasticization has become an effective method for molding polymer micro-parts due to its high efficiency, high precision, and low consumption, and has broad application prospects. In this paper, the flow properties of polymer melt were investigated for ultrasonic microinjection molding, using spiral flow testing under micro-scale. The experimental results show that it is feasible to test the fluidity of ultrasonic-plasticized polymers by using an Archimedes spiral mold with microchannels, which can directly reflect the filling performance of the melt in the micro flow channel. It was found that an increase of the ultrasonic amplitude, ultrasonic action time, plasticizing pressure, and mold temperature could effectively improve the polymer melt flow performance and increase the filling length. The former three parameters can increase the heat generation, and the mold temperature can slow down the melt cooling rate. Under the same conditions, the filling length of three polymers could be sequenced as *L*_PA66_ > *L*_PP_ > *L*_PMMA_. The influence of holding time and holding pressure on the filling length was smaller than that of the aforementioned four parameters. By reducing the channel thickness, the filling length was significantly decreased. However, the flow ratio of the polymer in the small-sized channel was no less than in the large-sized channel, which proves that the ultrasonic microinjection molding has an obvious advantage in filling micro-cavities with high aspect ratios.

## Figures and Tables

**Figure 1 polymers-11-00357-f001:**
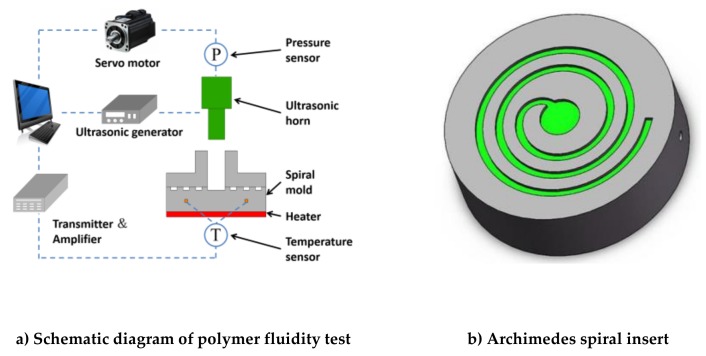
Ultrasonic microinjection molding equipment.

**Figure 2 polymers-11-00357-f002:**
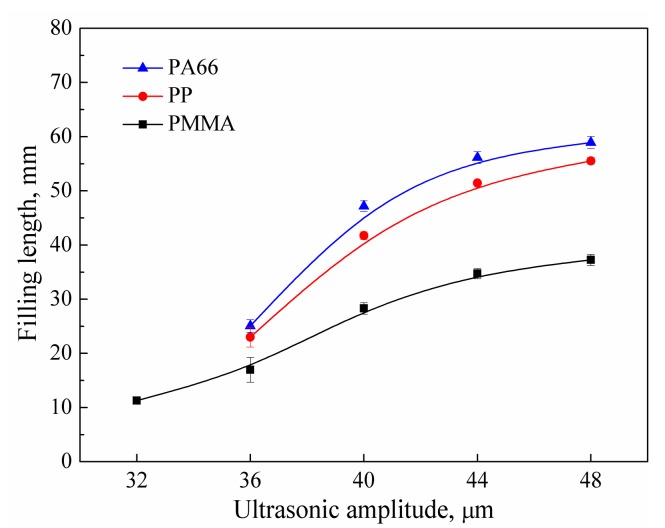
Influence of ultrasonic amplitude on the filling length of polymer melt (UT = 6 s, PPe = 14 MPa, HT = 6 s, HP = 14 MPa, MT = 60 °C; Mold I).

**Figure 3 polymers-11-00357-f003:**
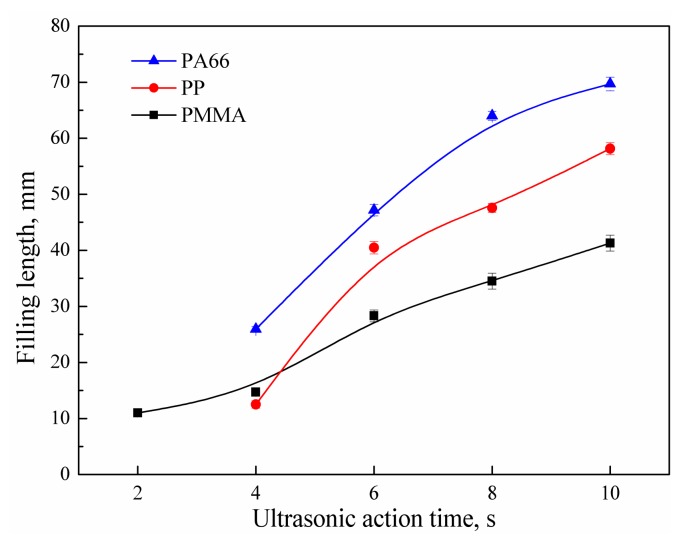
Influence of ultrasonic action time on the filling length of polymer melt (UA = 40 μm, PPe = 14 MPa, HT = 6 s, HP = 14 MPa, MT = 60 ℃; Mold I).

**Figure 4 polymers-11-00357-f004:**
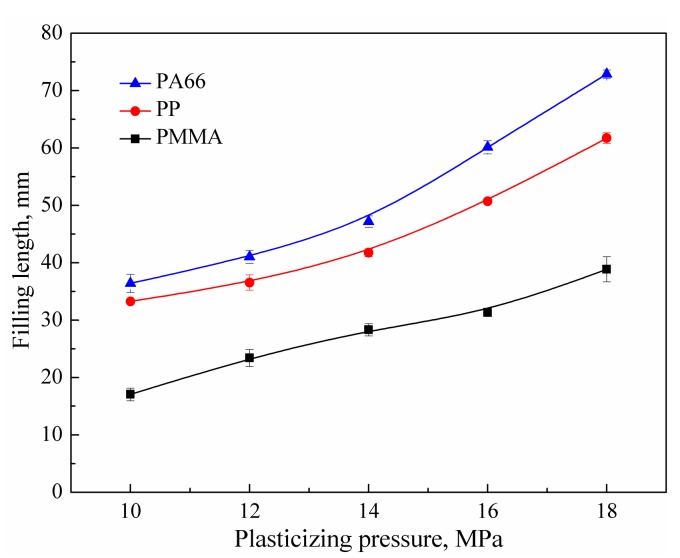
Effects of plasticizing pressure on the filling length of polymer melt (UA = 40 μm, UT = 6 s, HT = 6 s, HP = 14 MPa, MT = 60 ℃; Mold I).

**Figure 5 polymers-11-00357-f005:**
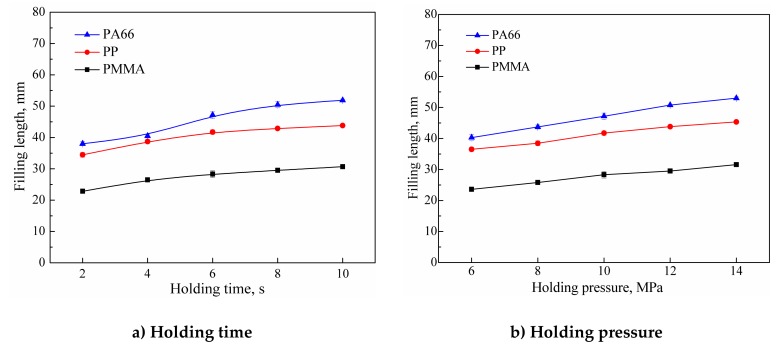
Effects of the holding process on the filling length of polymer melt (UA = 40 μm, UT = 6 s, PPe = 14 MPa, MT = 60 ℃; Mold I).

**Figure 6 polymers-11-00357-f006:**
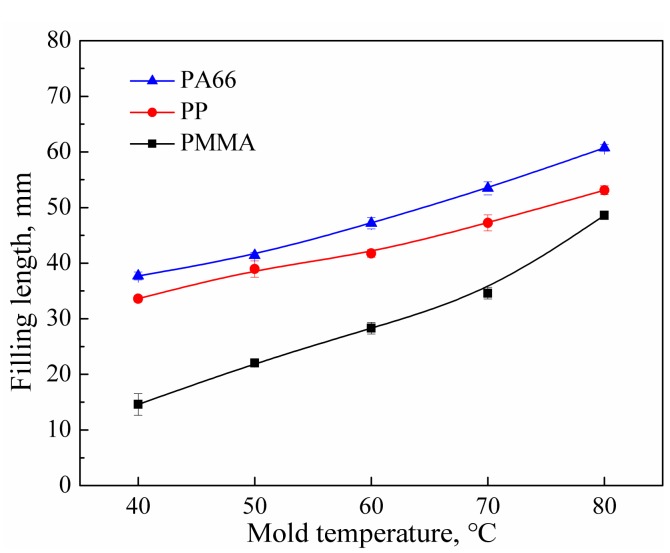
Effect of mold temperature on the filling length of polymer melt (UA = 40 μm, PPe = 14 MPa, UT = 6 s, HT = 6 s, HP = 14 MPa; Mold I).

**Figure 7 polymers-11-00357-f007:**
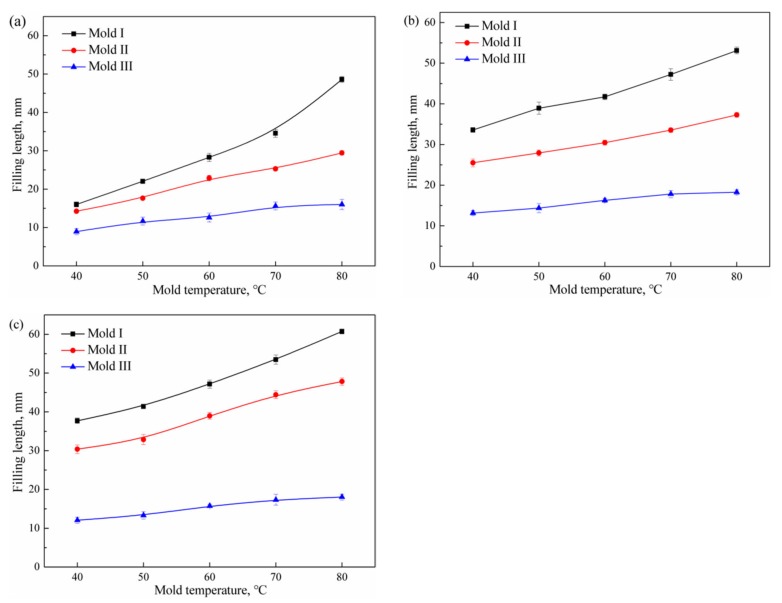
Filling length of polymer melt in different mold (**a**) PMMA; (**b**) PP; (**c**) PA66.

**Figure 8 polymers-11-00357-f008:**
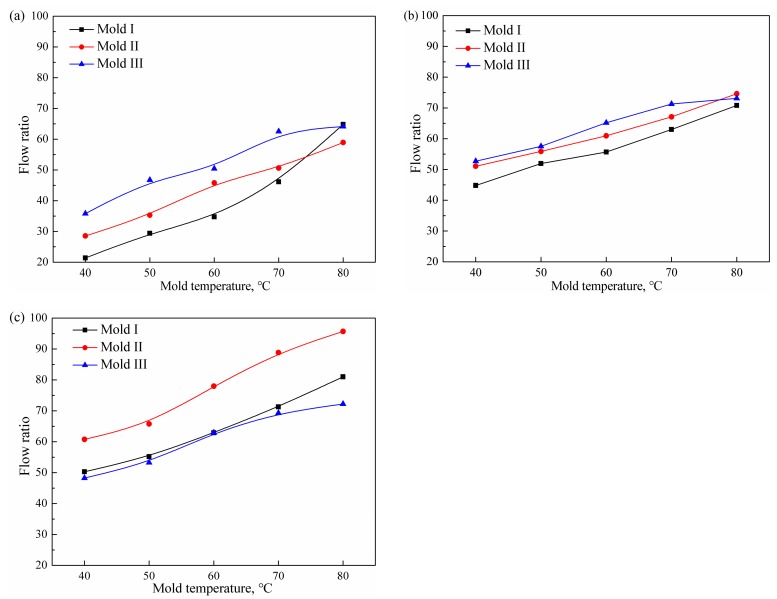
Flow ratios of polymers in different molds (**a**) PMMA; (**b**) PP; (**c**) PA66.

**Table 1 polymers-11-00357-t001:** Material properties of the investigated materials. PA66: polyamide 66; PMMA: polymethyl methacrylate; PP: polypropylene.

Material	Type	Density (g/cm^3^)	Melt Flow Index (g/10 min, ASTM D 1238)	Acoustic Impedance (Pa·s/m)	
				Solid	Liquid
PMMA	Amorphous	1.19	10	3.20 × 10^6^	1.23 × 10^6^
PP	Semi-crystalline	0.9	15	1.11 × 10^6^	4.28 × 10^5^
PA66	Crystalline	1.14	24	2.90 × 10^5^	1.10 × 10^5^

**Table 2 polymers-11-00357-t002:** Investigated process parameters and their values.

	Values	Value 1	Value 2	Value 3	Value 4	Value 5
Process Parameters						
Ultrasonic amplitude (UA)/μm		32	36	40	44	48
Ultrasonic action time (UT)/s		2	4	6	8	10
Plasticization pressure (PPe)/MPa		10	12	14	16	18
Holding time (HT)/s		2	4	6	8	10
Holding pressure (HP)/MPa		10	12	14	16	18
Mold temperature (MT)/°C		40	50	60	70	80

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
