# Peer review of "Characterization of the Fluidity of the Ultrasonic Plasticized Polymer Melt by Spiral Flow Testing under Micro-Scale"

_polymers, 2019, doi:10.3390/polym11020357_

Round 1

Reviewer 1 Report

The manuscript “Characterisation of the fluidity o the ultrasonic plasticized polymer melt by spiral flow testing under microscale” by Bingyan Jiang et. al., is an interesting paper where the authors investigate The fluidity of the molten polymer plasticized by ultrasonic vibration using an Archimede spiral mold.

The manuscript is well written and well-articulated, and the single factor analysis approach was used for determining the role of the process parameters.

I recommend the article to be published after minor revisions:

Title: line 2 “ultrasonic”

Line 34: “have report”

Line 67: “an reference”

Iine 71: “there are overall three kinds of material was used …”

Author Response

Dear Reviewer:

Thank you for your review of our manuscript (ID: polymers-436797). We appreciate the concerns and suggestions provided and have revised our manuscript accordingly: 

Point 1: Title: line 2 “ultrasonic”. 

Response 1: Replace “ultraosnic” with “ultrasonic”.

Point 2: Line 34: “have report”.

Response 2: Replace “have report” with “have reported”.

Point 3: Line 67: “an reference”.

Response 3: Replace “an reference” with “a reference”.

Point 4: Line 71: “there are overall three kinds of material was used …”

Response 4: Replace “There are overall three kinds of material was used in this study i.e.” with “There are three kinds of material used in this study, i.e.,”.

With sincere thanks

Reviewer 2 Report

I have only one comment. I didn't found the channel width of the spiral. It is also the important data for understanding of the cavity filling proces. Please add that information.

Author Response

Dear Reviewer:

Thank you for your review of our manuscript (ID: polymers-436797). We appreciate the concerns and suggestions provided and have revised our manuscript accordingly: 

Point 1: I have only one comment. I didn't find the channel width of the spiral. It is also the important data for understanding of the cavity filling process. Please add that information. 

Response 1: Channel width is 1500 μm and has been added to line 98.

With sincere thanks
